# Insights into Repeated Renal Injury Using RNA-Seq with Two New RPTEC Cell Lines

**DOI:** 10.3390/ijms241814228

**Published:** 2023-09-18

**Authors:** B. Alex Merrick, Negin P. Martin, Ashley M. Brooks, Julie F. Foley, Paul E. Dunlap, Sreenivasa Ramaiahgari, Rick D. Fannin, Kevin E. Gerrish

**Affiliations:** 1Mechanistic Toxicology Branch, Division of Translational Toxicology, National Institute of Environmental Health Sciences, Research Triangle Park, NC 27709, USA; foley1@niehs.nih.gov (J.F.F.); dunlapp@niehs.nih.gov (P.E.D.); ramaiahgari.sreenivasa@epa.gov (S.R.); 2Viral Vector Core, Neurobiology Laboratory, Division of Intramural Research, National Institute of Environmental Health Sciences, Research Triangle Park, NC 27709, USA; martin12@niehs.nih.gov; 3Biostatistics and Computational Biology Branch, Integrative Bioinformatics Support Group, Division of Intramural Research, National Institute of Environmental Health Sciences, Research Triangle Park, NC 27709, USA; ashley.brooks@nih.gov; 4Molecular Genomics Core Laboratory, Division of Intramural Research, National Institute of Environmental Health Sciences, Research Triangle Park, NC 27709, USA; fannin@niehs.nih.gov (R.D.F.);

**Keywords:** kidney, renal proximal tubule, cisplatin, nephrotoxicity, aflatoxin B1, transcriptomics, RNA-seq, immortalization, nephrogenesis

## Abstract

Renal proximal tubule epithelial cells (RPTECs) are a primary site for kidney injury. We created two RPTEC lines from CD-1 mice immortalized with hTERT (human telomerase reverse transcriptase) or SV40 LgT antigen (Simian Virus 40 Large T antigen). Our hypothesis was that low-level, repeated exposure to subcytotoxic levels of 0.25–2.5 μM cisplatin (CisPt) or 12.5–100 μM aflatoxin B1 (AFB1) would activate distinctive genes and pathways in these two differently immortalized cell lines. RNA-seq showed only LgT cells responded to AFB1 with 1139 differentially expressed genes (DEGs) at 72 h. The data suggested that AFB1 had direct nephrotoxic properties on the LgT cells. However, both the cell lines responded to 2.5 μM CisPt from 3 to 96 h expressing 2000–5000 total DEGs. For CisPt, the findings indicated a coordinated transcriptional program of injury signals and repair from the expression of immune receptors with cytokine and chemokine secretion for leukocyte recruitment; robust expression of synaptic and substrate adhesion molecules (SAMs) facilitating the expression of neural and hormonal receptors, ion channels/transporters, and trophic factors; and the expression of nephrogenesis transcription factors. Pathway analysis supported the concept of a renal repair transcriptome. In summary, these cell lines provide in vitro models for the improved understanding of repeated renal injury and repair mechanisms. High-throughput screening against toxicant libraries should provide a wider perspective of their capabilities in nephrotoxicity.

## 1. Introduction

Proximal tubule cells of the kidney are primary sites for reabsorption of ions and intermediary metabolites driven by energy-driven transporters. Renal proximal tubule epithelial cells (RPTECs) express multiple transporter systems that have broad substrate specificity and special functions including the disposition of many xenobiotics [1,2]. While RPTEC oxidative and conjugation enzymes have the capacity to metabolize and dispose of foreign substances, there is the potential for chemical-induced injury to this important site within the kidney nephron. Functional capacity and reserve of the kidney is substantial but over time, repeated injury can eventually cause clinical symptomology and evidence of injury observed by biomarkers in urine and serum [3]. Although the causes of chronic kidney disease (CKD) vary, the insidious nature of long-term damage prior to detection often leave clinicians in a difficult position for therapeutic treatment and reversal.

CKD can occur after long-term renal injury as a comorbidity from diabetes and hypertension [4], by environmental toxicant exposures [5,6], or as a side-effect of therapeutic treatment [7]. Cisplatin (dichlorodiamine platinum) is a widely used chemotherapeutic agent for treating a variety of solid malignant tumors. However, over time cisplatin causes nephrotoxicity targeting renal proximal epithelial cells that is produced by a complex process of oxidative stress, inflammation, and apoptosis [8] that remains incompletely understood. Research suggests the pathogenesis of CKD produced by chronic cisplatin exposure models is likely different from acute kidney injury by cisplatin [9]. Some CKD mouse models have focused on episodic or repeated, low-level kidney injury as one path to transitioning to CKD [10]. However, other studies have used repeated sub-cytotoxic exposure regimens of cisplatin to better represent the cellular processes leading to CKD produced in animal models [10,11] as well as in vitro models of 2D human cell cultures [12,13], microfluidic devices [14], or kidney organoids [15]. For example, human TERT1-RPTEC were treated with subcytotoxic levels of cisplatin for 2 weeks and using a multi-omics approach, investigators reported activation of several kidney injury pathways [13]. Another study that used stem cell-derived human kidney organoids found that an alternate day exposure at a subcytotoxic concentration of 5 µM cisplatin for one week resulted in increased cytokine release into the culture media and gene expression analysis that supported activation of the TNF-α signaling pathway [15].

Human RPTECs immortalized by hTERT (telomerase reverse transcriptase) have proven an extremely useful model for in vitro studies [16,17,18]. Other human RPTEC lines have been created for experimental use such as HK-2 cells [19] immortalized by transduction with HPV16 E6/E7 genes; TH1 cells [20] immortalized with SV40Tag and hTERT; and SA7K cells [21] immortalized by nucleotransfection of zinc finger nuclease (ZFN) pairs. There is also a need for animal RPTEC models whose in vitro results can be translatable into experimental animal systems. Proximal tubule cells from various mouse strains have been immortalized by transfection of SV40-LgT antigen containing vectors [22,23,24,25] or by isolation of renal proximal tubules from transgenic mice expressing SV40-LgT antigen [26,27,28,29,30] and p53 KO mice [31]. To our knowledge, there is not an hTERT immortalized mouse RPTEC line.

We have been intrigued by the experimental approaches using low-level, repeated exposure [11,13,15] that could lead to improved modeling of chronic kidney disease. Our interests are in better defining the activation of molecular pathways, concentration-responsive transcripts, and new biomarkers in chronic kidney injury. We hypothesized that varying the immortalization methods starting from the same primary proximal tubule cells could produce cell lines that respond with distinct genes and pathways after exposure to different nephrotoxicants. Towards this aim, we created two new cell lines immortalized in vitro with a lentivirus vector carrying SV40 Large T antigen (LgT) or a lentivirus vector with human TERT (hTERT). In this report, we studied two nephrotoxicants, cisplatin (CisPt) and aflatoxin B1 (AFB1), at varying exposure times and concentrations using RNA-seq in these two new mouse RPTEC lines. The data showed that repeated exposure with CisPt (commonly used to induce renal injury) and AFB1 (a milder kidney toxicant) produced unique transcriptional responses to proximal tubular injury that could be further exploited in mechanistic studies and gene-specific mouse models.

## 2. Results

Figure 1 shows the epithelial morphology of confluent primary moRPTEC (mouse renal proximal tubule epithelial cell) cultures and immortalized moRPTEC lines. The morphology changed with increasing cell density. At a lower density, all the cells were spindle-shaped often with multiple thin projections. However, at confluence the cells took on an irregular polygonal shape arranged in a compact cobblestone-like appearance. The immortalized cells were generally similar in morphological appearance to primary RPTEC cultures, and all the cells had well-defined nuclei and polygonal cell membrane boundaries. The inclusion of either neomycin or puromycin in the culture medium produced a continued selection of cells expressing immortalization vectors in hTERT or LgT lines, respectively.

An initial experiment was conducted to test for the concentration-dependent transcriptional responses to repeated sub-cytotoxic exposures to nephrotoxicants, CisPt, or AFB1 using RNA-seq. We chose a 72 h time point to allow for the cumulative effects of repeated chemical exposure. RNA-seq results are in Appendix A, provided in data tabs for each concentration of CisPt and AFB1, and for each cell line, culminating in a total of over one million data points. Each tab includes data for 55,402 ENSEMBL transcripts that includes the gene symbol and name, RefSeq and Entrez IDs, fold change from control, and standard error and *p* values. Using DESeq2 for differential expression analysis to CisPt and AFB1, we filtered for two-fold expression differences at an adjusted *p* ≤ 0.05 value to diminish false positives. However, for PCA analysis all differentially expressed genes were used to optimally visualize the separation among the treatment groups. In the absence of cell death, we observed conventional biomarkers of renal injury, such as *Havcr1* (Kim-1) and *Il6*, were mildly increased, over concentration and time with AFB1 and CisPt. For example, *Havcr1* ranged from a 2- to 4-fold increase in either cell line with either chemical treatment at 72–96 h. In addition, we often found increased expression of *Mmp9*, *Mmp10*, *Mmp13*, and *Mmp25* that was consistent with their recognized roles in renal damage [32,33,34,35]. (Please note all gene names are defined in the Abbreviations section.)

Figure 2 shows that CisPt readily produced the strongest transcriptional response in both cell types by the number of up- and down-regulated DEGs (2X fold change, pAdj ≤ 0.05) compared to AFB1. For example, at the highest CisPt concentration at 2.5 µM there were 2000–5000 total DEGs observed in each cell line. By comparison, LgT produced slightly more than 1139 total DEGs at the highest AFB1 concentration of 100 µM. hTERT moRPTECs were significantly less responsive to AFB1, showing only five DEGs or less, over a concentration range of 12.5–100 µM. DEGs by cell type and concentration of AFB1 and CisPt are provided in Appendix A.

The AFB1 expression response was most pronounced in LgT moRPTECs compared to hTERT cells. LgT cells showed no visible toxicity and minor expression increases in *Havcr1* (Kim1), *Mmp10*, and *Mmp25* from repeated 12.5–100 µM AFB1 exposure after 72 h. However, several AFB1-induced DEGs in the LgT cells showed concentration-related upregulation in expression at 10-fold or greater (Appendix A). Upregulated genes included transcription factors containing homeobox domains (*Barx1*, *Gbx2*, *Nkx2-9*, *Tlx3*) or bHLH domains (*Hes2*, *Tfap2e*), as well as transcripts with immune functions including *Il6* and the chemokine receptor, *Cxcr4*. In addition, pathway analysis of transcripts with increased expression showed activation of CREB, S100 family, and GPCR (G-protein coupled receptor) signaling pathways and phagosome/lysosome formation. The relative sensitivity of LgT cells and lack of response in hTERT cells was a distinguishing feature of these two immortalized moRPTEC types.

Principal Component Analysis (PCA) provides a useful unsupervised method to visualize similarities and differences contained within the large transcriptomic datasets [36] from the two cell types, two chemical treatments, and five concentrations. PCA analysis was performed on normalized expression values of all genes in each cell type and are shown in Figure 3. Transcriptional responses show that each cell type responded to each chemical treatment with a similar transcriptional profile, such that the representation of the LgT cell responses clustered closer to each other (left side of plot) and the hTERT responses clustered nearer to each other (right side of plot). The comparatively milder transcriptional response to AFB1 was reflected by the closer clustering of each concentration group within each cell type (blue for hTERT, AFB1; and red for LgT, AFB1). Compared to AFB1, the CisPt treatment produced a larger separation of expression responses among increasing concentration groups in each cell line (green for LgT; yellow for hTERT) without cytotoxicity up to 2.5 µM. However, at levels greater than 2.5 μM CisPt, visible morphologic changes occurred, including cell rounding, detachment, and cell death.

The greater responsiveness of moRPTEC lines to CisPt compared to AFB1 motivated us to strengthen our focus on transcriptional responses to CisPt. The top one hundred overexpressed transcripts were reviewed for concentration-related responses to CisPt, based on increasing fold change with increasing concentration. As shown in Figure 4, there were 19 transcripts in the LgT cells and 14 transcripts in the hTERT cells that showed a general responsiveness to increasing CisPt concentrations. The highest fold transcript changes observed for LgT were *Hoxc12*, *IL18*, and *Msx1* (see upper inset) and the other 16 transcripts were increased >100-fold at the highest CisPt concentration of 2.5 µM. Transcript fold changes were less pronounced in the hTERT cells than in the LgT cells but still were almost 40- to 100-fold at the highest CisPt concentration. Some transcripts were slightly decreased at lower concentrations before showing a rise in the fold change with increasing concentration. Physiological functions of concentration-related transcripts in both cell lines involved kinases, the immune system, growth and development, ion channels, and cell signaling.

Activation of canonical pathways was determined by considering the highest 1000 DEGs and the lowest expressing 1000 DEGs as inputs into an Ingenuity Pathway Analysis® (IPA) analysis platform that ranks known canonical pathways according to the fold changes and proportion of DEGs populating each pathway. The results in Figure 5 show activation of various biochemical and signaling pathways for which the top scores were notably shared by both cell types. Shared pathways between LgT and hTERT cells included CREB signaling in neurons (cellular plasticity), G-Protein receptor coupled signaling (signal transduction), phagosome formation (tissue remodeling and inflammation), and breast cancer regulation by *Stathmin 1* (nucleus microtubule dynamics). Other pathways of interest involved immunoregulatory and immune cell recruitment, and also regulation of growth, differentiation, and developmental processes. In addition, the highest 1000 DEGs were examined for a common expression response in hTERT or LgT cells. We found only about 5% of known annotated DEGs were identically shared between the two cell types, though differential expression of homologous gene family members was frequently observed.

We conducted a second experiment on a CisPt time course study to test for similarities and differences in the transcriptional response to 2.5 µM CisPt between the hTERT and LgT cell lines. Our preliminary study (see Figure 2) found that 2.5 µM CisPt was the maximum concentration without cell death in both the cell types for producing a robust transcriptional response. The shortest time points in the time course experiment were single exposures after 3 and 6 h and then at 24 h. Thereafter, repeated daily CisPt exposures and cell harvesting were conducted after 48, 72, and 96 h (see Appendix A for treatment regimen). Each CisPt time point was compared to its own untreated control with all exposures performed in triplicate. RNA-seq analysis was performed on each well for these samples.

Fold-change data of all the characterized transcripts at each time point were cataloged in Appendix A and were plotted by PCA (Figure 6) to determine differences to exposure time and cell type. The results showed the hTERT transcriptional responses appeared earlier than the LgT, and the responses of the two cell lines began to noticeably diverge after 24 h of CisPt treatment. Accordingly, the transcriptional responses between hTERT and LgT widened from 48 to 96 h after repeated CisPt exposure. Pathway analysis was conducted on the DEGs over the time course of CisPt treatment (Appendix A), but notably only 24 h, 48 h, 72 h, and 96 h time points had sufficient numbers of DEGs to populate the pathways in this experiment. In accordance with our previous data examining changes in concentration, CREB signaling in neurons and G-protein coupled receptors pathways changed over the exposure time for both the cell lines. In addition, the S100 family signaling pathway (immune and Ca^++^ signaling) and axonal guidance signaling (cell migration and progenitor dynamics), were also shared signaling pathways over the exposure time.

Figure 7A shows that the number of DEGs (*p* ≤ 0.05, 2X fold change) over time after CisPt treatment climbed steadily in both the cell lines after 24 h, with up-regulated transcripts outpacing the number of down-regulated transcripts at each time point (Appendix A). By 72 h, there were 2000–3000 DEGs, and by 96 h there were almost 4000 total DEGs in both the cell lines (Figure 7B).

Most DEGs had annotated identities and functions (Figure 7B). Annotated transcripts were used in Venn diagram analysis to determine shared and unique DEGs and to explore canonical pathways for each cell line in their temporal responses to CisPt. Figure 7C shows that 602 DEGs were shared among the hTERT cells or 14.9% of the 4043 total DEGs at 48–96 h. For the LgT cells, there were 1118 shared DEGs or 24.4% of the total DEGs at 48–96 h. Canonical pathways populated with our study’s DEGs using IPA software (Version 94302991, Release date 27 May 2023) provided further insight into the CisPt effects. We initially focused on the higher numbers of DEGs in the 48–96 h datasets for pathway analysis in each cell line in Figure 7C that is further detailed in Appendix A. The 602 common DEGs in the hTERT cells from 48 to 96 h suggested activation of the S100 Family, eicosanoid signaling, phagosome formation, GPCR signaling, and GPCR-mediated enteroendocrine signaling. The 1118 common DEGs in the LgT cells from 48 to 96 h similarly suggested activation of S100 Family and GPCR-mediated enteroendocrine signaling, CREB signaling, axonal guidance, and cardiac hypertrophy signaling. Activation of unique pathways in each cell type varied somewhat at each time point. For the hTERT cells, engagement of calcium signaling, immune response, and neurotransmitter pathways (e.g., glutaminergic GABA signaling) were notable. For the LgT cells, activation of xenobiotic (e.g., GSH detoxification, *Ahr* signaling) and various immune pathways (e.g., Pathogen-induced cytokine storm, Airway pathology in COPD, wound healing) were observed.

Comparisons of the pathway responses in the hTERT versus LgT cell lines were also made at each CisPt time point. At 24–48 h (see data in Appendix A), unique pathways like Netrin (axon guidance) and eNOS signaling for hTERT were observed, and the unique pathways activated in the LgT cells involved engagement of the Serotonin receptor and wound healing pathways. For pathways common to the two cell types at 24 h, Phosphoinositide signaling and Triacylglyerol degradation pathways were found. At 48 h, breast cancer regulation by *Stathmin 1* (microtubule regulation) and factors promoting cardiogenesis in vertebrates (*Wnt*/*Bmp* axis) occurred in addition to other immune and neural pathways.

We note that most differential expression occurred at 72 and 96 h as indicated in Figure 7D where one quarter of the DEGs were shared among the hTERT and LgT cells at 72 h (23.9% shared) and at 96 h (26.7% shared). Several shared DEGs between the hTERT and LgT cells are related to kidney function including KCN-family potassium channel transcripts (12 DEGs at 72 h, 10 DEGs at 96 h), *SCN*-family sodium channel transcripts (4 DEGs at 72 h, 5 DEGs at 96 h), as well as *SLC*-family ion and drug transporters (18 DEGs at 72 h, 40 DEGs at 96 h). For example, both the hTERT and LgT cells showed increased expression in the potassium voltage-gated channel, *Kcnd3*, at 10–14-fold in the hTERT and 18–183-fold increase in the LgT cells at 72 h and 96 h, respectively.

Common and unique DEGs at 72 and 96 h in Figure 7D for the two cell types were analyzed for canonical pathway activation. Common pathways for both cell types involved signaling of the S100 family proteins, axonal guidance, serotonin receptor, cardiac hypertrophy, CREB transcription factor, and enteroendocrine systems. The unique pathways to each cell type involved immune, neurotransmitter, and repair pathways as further described in Appendix A.

We were also interested in the similarity and diversity of the most pronounced gene expression changes over time. The 20 transcripts with the greatest fold-change expression from 24 h to 96 h are shown in Table 1. The 20 transcripts with the lowest fold-change are provided in Table 2. Few transcriptomic changes occurred at 3 and 6 h, so these time points were not considered. Red (Table 1) or green (Table 2) highlighted DEGs show two or more occurrences in one cell line, often at adjacent time points. Orange highlighted DEGs show two or more occurrences found in both cell lines. Upregulated DEGs common to both cell types in Table 1 involved synaptic adhesion/organizing molecules [37] (SAMs) such as *Cbln2*, *Sparcl1*, and *Otof*, and also tissue remodeling proteins including *Ctsq*, *Ces2e,f* and *Mpped1*, and ion channel components like *Scn4b*. Common down-regulated DEGs in both cells in Table 2 involved intracellular intermediary metabolism and signaling proteins such as *Fhit* (purine metabolism), *Inpp4b* (Pi3K/Akt signaling), *Plcb1*, and *Msra* (signal transduction), and adjustments in SAM proteins like *Tenm4* and *Cdh13*, as well as transcription factors for differentiation (*Sox5*).

A heat map in Figure 8A was constructed to see if the concentration-related transcripts we proposed from Figure 4 could be reproduced over the 3 h–96 h CisPt time course. Results show most transcript expressions (25 of 33) did increase over time in one or both cell lines with the highest changes occurring at 72 h and 96 h. However, some transcript changes were cell line specific. For the hTERT cells, *Cacna1i*, *Fbn1*, *Flt3*, *Hc*, *Lhx2*, and *Tmem163* were increased but with minimal expression changes in the LgT cells. For the LgT cells, *Adcyap1*, *Msx2*, and *Slitrk2* increased while lower expressions of these transcripts were observed in the hTERT cells. Fold changes and mean base counts (number of reads as a relative measure of transcript abundance) are presented in Appendix A for each transcript.

Similarly, Figure 8B shows an expression heatmap of the fold changes for regulatory transcripts that were upstream of the canonical pathways. These upstream regulatory transcripts were consistently increased over time in one or both cell lines.

We found these “upstream regulatory transcripts” by IPA Core analysis that predicts known upstream pathway regulators by populating biochemical pathways with DEGs from our RNA-seq data. A list of predicted upstream regulatory transcripts responding to CisPt was assembled in Appendix A by fold change and their mean base counts (number of reads as a relative measure of transcript abundance). Such upstream regulatory transcripts included components in inflammation and immune recruitment (e.g., interleukins, *S100a8*/*S100a9*; *Ltb4r1,2*), growth factors (e.g., *Csf2* and *Fgf3*); promoters of cell morphogenesis (e.g., *Wnt11*, *Nog*), remodeling (e.g., *Mmp12*), and differentiation (e.g., *Neurod1*, *Trp63*, *Trp73*). We note that fold changes for *Trp53* were included in Figure 8B for comparison to the changes in *Trp63* and *Trp73*, but *Trp53* expression was relatively unaffected in both the cell types.

We performed pathway analysis for 24, 48, 72, and 96 h time points (there were insufficient DEGs at 3 and 6 h) to determine either activating or inhibitory pathway engagement. These results are summarized in Appendix A. For example, *Csf2* (colony stimulating factor 2) is a hematopoietic growth factor that acts as an activating regulator while *Inf4* (interferon regulatory factor 4) negatively regulates TRL (Toll-like receptor) signaling in innate and adaptive immune systems. The numbers of activated or inhibited pathways affected by upstream regulatory transcripts are shown in Figure 9. Overall, there was a 3:1 greater number of predicted activating vs inhibitory pathways, which were more prevalent at the 72 and 96 h time points.

For the purpose of confirming the direction of the fold change from the RNA-seq data, we analyzed select transcripts for differential expression by CisPt compared to the untreated control using a multiplexed probe hybridization platform. Differential expression of several transcripts using nCounter® (Nanostring, Seattle, WA, USA) is shown in Figure 10. Log2 transformed expression values were used to better visualize the transcript changes on the same plot. We found low expression transcripts with fewer RNA-seq reads were more challenging to validate such as *Adcy8*, *Adcyab1*, *Fbn1* and *Flt3*, and *Slitrk5*. However, the transcripts with either high differential expression by RNA-seq (e.g., *Nog* and *Prl2a1*) or modest differential expression by RNA-seq but high read counts like *Cdkn1a*, *Trp53*, and *Mdm2* demonstrated significant increased expression by the nCounter platform in both the cell lines. Similarly, for high read transcripts like *Gadd45*, reduced expression of this transcript was observed in agreement with the RNA-seq results.

## 3. Discussion

Lentivirus immortalization was used to establish two new moRPTEC lines by lentivirus vectors containing SV40-LgT Ag and hTERT. Short tandem repeat (STR) profiling confirmed these immortalized cell types were murine lines. We tested two well-known nephrotoxicants, CisPt and AFB1, to distinguish the expression characteristics of each cell type. AFB1 generally requires metabolic activation to the 8,9-epoxide to produce toxicity by DNA and protein adducts [38] making liver a primary target organ, although AFB1 adducts have also been documented in the kidneys of rats and mice [39]. Here, only the LgT moRPTECs showed substantial transcriptional changes (1139 DEGs) in response to AFB1 at 72 h. Upregulation of typical toxicity markers like Kim1 (*Havcr1*), *Il6*, and *Mmp*’s were observed with only minor increases in the LgT cells after 72 h. However, AFB1-mediated increases by concentration for *Cxcr4* [40,41] and *Il6* [42] are consistent markers of kidney injury. Such changes could be responsible for the induction of multiple developmental transcription factors [43,44] such as *Barx1*, *Gbx2*, *Nkx2-9*, *Tlx3*, *Tfap2e*, and *Hoxc12*. Activated pathways across all the AFB1 concentrations included CREB signaling, phagocytic activity, S100 family signaling, and GPCR signaling. Activation of these pathways likely reflects an intrinsic renal cellular repair program [45]. Although some research suggests AFB1 produces oxidative stress that underlies renal injury [46], others view renal effects as an ill-defined consequence of liver AFB1 metabolism and downstream toxicity [47]. The exact mechanism of AFB1 effects in the LgT moRPTECs is not known. However, recent studies in AFB1-exposed human populations suffering from early-stage renal damage [48,49] highlight the need for new in vitro models to study direct AFB1 contributions to kidney damage leading to diminished function and disease. (Please note all gene names are defined in the Abbreviations section.)

CisPt nephrotoxicity represents a complex series of events including injury to the proximal tubule epithelia and renal vasculature, immune response, and disrupted cellular redox potential [50]. Cultured cell models provide controlled conditions to study the molecular changes underlying proximal tubule injury. We examined expression changes over concentrations that showed similar trends in a time course experiment. Elevation in nephrogenesis hox genes, notably *Hoxc12* and *Hoxc13*, are consistent with other reports describing Hox upregulation during long-term, pathologic kidney remodeling [51]. Increased *Il6* and *Il18* are known cytokines in acute renal injury [52]; we also observed increased expression of several other transcripts contributing to a renal immune response after CisPt, including *Itk* [53], *Ptafr* [54], *Selp* [55], *Hc* [56], and *Nlrp12* [57]. It was also very interesting to find upregulation of several neural-related transcripts over time that included *Nog*, *Negr1*, *Lhx2*, *Ntrk1*, *Scn7a*, *Slitrk2*, *Slitrk5*, and *Cacna1i*. We would argue these genes play roles in regenerative processes in response to tubular epithelia injury. In fact, similarities between the renal and neural pathway circuits have been documented. For example, Noggin (*Nog*) is a secreted polypeptide important in the early development of nerve, muscle, and bone tissue. Upregulation of Noggin has also been found as a critical factor in kidney regeneration in a rodent model of ischemia-induced acute renal failure [58]. Similarly, *Lhx2* [59], *Scn7a* [60], *Negr1* [61], *Ntrik1* [62], and *Grm1* [63] are increased during injury and repair in various tissues in addition to neural cell types. Upregulation of several of these transcripts was also found with the Nanostring multiplex platform.

Another aspect to CisPt toxicity is its pharmacokinetic entry into and removal from proximal tubule cells. Entry can occur by passive diffusion as well as by facilitated movement from specific membrane transporters [64,65]. Clinically, the basement membrane organic cation transporter, *Slc22a2* (Oct2), is well recognized as highly responsible for CisPt bioaccumulation over the course of chemotherapeutic treatment resulting in nephrotoxicity. Since *Slc22a2* is poorly expressed in human and rodent immortalized RPTEC lines, many studies have used Oct2-overexpression systems to study its transporter functions with various therapeutics [66]. In murine proximal tubules, we note that other basolateral transporters like Ctr1 (*Slc31a1*) are highly expressed in our cell lines and have a demonstrated function in CisPt uptake in other studies [67]. Although we have not yet performed functional assessment for CisPt transport, several transport-capable proteins for CisPt are already well expressed in LgT and hTERT moRPTECs including *Slc31a1*, *Slc31a2* (e.g., copper transporters Ctr1 and Ctr2), and *Slc47a1* (e.g., *Mate1*; multidrug and toxin extrusion protein 1).

Fold changes of the top 20 DEGs reflect response intensity of individual transcripts to CisPt, though just a small proportion of the same DEGs were conserved in the top DEG group over 24–96 h. We also found several transcripts that were shared between the two cell lines (e.g., ↑ *Otof*, ↑ *Scn4b*, ↓ *Fhit*, ↓ *Pacrg*) as well as observing some unique high-expression and low-expression transcripts seen either in the hTERT (e.g., ↑ *Ikzf3*, ↓ *Umod*) or in the LgT cells (e.g., ↑ *Dynap*, ↓ *Clstn2*). These observations reflect the diversity of biological responses at the individual gene level as cells adjust to repeated CisPt exposure over time. In addition, pathway-level interrogation is helpful for interpreting differential expression of large gene sets [68]. Prior investigations reported engagement of several pathways during CisPt toxicity. CREB-mediated activation of the EGFR-Ras/ERK pathway was reported as a means to overcome H_2_O_2_ stress from 25 µM CisPt for up to 24 h exposure in mouse proximal tubule cells [69], or apoptosis through the ERK-p66shc pathway [70]. Another study using a multi-omic approach exposed RPTEC/TERT1 human cells to subcytotoxic concentrations of CisPt (0.5 and 2 µM) repeatedly over a course of 2 weeks and demonstrated the involvement of *Trp53* signaling, *Nrf2* and mitochondria-mediated oxidative stress, as well as mTOR, AMPK, and EIF2 signaling pathways [13].

Interestingly in our studies, we did not observe significant fold changes (≤1.1X) in the *Trp53* transcript by RNA-seq in either the LgT or hTERT cell lines over concentration and time. Immortalization by SV40 Large T antigen’s posttranscriptional effects involve binding *Trp53* and *Rb*-family tumor suppressors [71]. Nominal increases in *Trp53* (≤1.3 fold) from Nanostring data agreed with the RNA-seq findings but were statistically significant due to reproducibility of high fluorescence counts. Similarly, the Nanostring data showed modest fold increases in *Cdkn1a* (p21) at 1.3X and 1.4X fold in the hTERT cells and 1.7X and 2.4X fold in the LgT cells at 72 h and 96 h, respectively. However, *Gadd45* (p53-dependent gene) was down-regulated. *Mdm2* levels shown by Nanostring were slightly increased in expression and ranged from 2 to 4 fold in both types. Although *Mdm2* and *Cdkn1a* are generally controlled by *Trp53* in primary cells, *Trp53*-independent roles for these transcripts are being increasingly recognized [72,73]. Furthermore, the absence of apoptosis-related DEGs and pathways in response to exposure over concentration and time suggests *Trp53*-mediated cell death did not play a role in the response of our cell lines to nephrotoxicants under our exposure conditions. However, we did note large fold changes in the *Trp53*-related family members, *Trp63* and *Trp73*, in both the cell types by RNA-seq. *Trp63* and *Trp73* have roles in the differentiation and development of squamous epithelial and neural cells, respectively [74], and recently have been found to play critical roles in kidney cell morphogenesis in MDCK cells [75,76]. An intriguing role for *Trp63* and *Trp73* in proximal tubule cell recovery after injury merits further investigation.

Pathway activation marked by the presence of DEGs are controlled by upstream regulators and compiling these cellular events could give broader insights into CisPt-induced effects. The number of activated pathways outnumbered the inhibitory pathways as they both increased over time from 24 to 96 h. We focused our attention on the upstream regulators that showed significant differential expression with the potential to activate or inhibit pathways. In our study, immune, neurogenic, and tissue remodeling and repair pathways were the primary cellular responses to subcytotoxic CisPt exposure in these two new cell lines. We observed *Il6* and *Il18* pathway engagement that was consistent with transcriptional increases over concentration and time in both the cell types. However, other critical immune transcripts and pathways were also highly upregulated including the *S100a8*/*S100a9* pathway noted in AKI [77]; cytokines *Il17f*, *Il24*, *Il33*, and *Tnf* pathways in kidney injury [78,79,80]; the arachidonic acid metabolite binding *Ltb4r1* and *Ltb4r2* pathway [81] and the *Lilrb4a,b* receptor pathway, both involved in leukocyte recruitment [82,83]; the chemoattractant *Cxcr4* [84] and pathway activation by the anaphylatoxin, *Hc* (C5a) [85]; and the prostanoid-producing *Ptgs2* transcript (Cox2) pathway [86]. Upregulations in *Wnt11* [87], *Fgf3* [88], and *Mmp12* [89] have been documented in repair processes accompanying kidney injury. *Csf2*, secreted from renal proximal tubules cells during AKI, can promote the transition of proinflammatory M1 macrophages to an anti-inflammatory M2 phenotype in a dose and time-dependent manner [90], contributing to injury repair. Here, we also observed *Csf2* pathway activation and increased expression over 24–96 h in both the cell lines. In addition, the GPCR-pathways were widely activated in this study since many GPCR ligands were likely elevated during injury that could include hormones, peptides, proteins such as chemokines, ions like Ca^++^, synaptic transmitters such as glutamine, and intermediary metabolites ranging from fatty acids, ATP, ADP, to many others [91].

The molecular pathogenesis of acute renal injury is highly complex but in vitro systems can help identify specific genes and orthologs for unraveling the multiple pathways involved for early detection, therapeutic intervention, and prevention. Instead of a single nephrotoxicant challenge, we feel advanced understanding in chronic kidney disease can be gained from repeated subcytotoxic exposures. We observed most expression changes at 72 h or 96 h but recognize some genes and pathways were activated as early as 24 or 48 h. We interpreted these temporal transcription differences in genes and pathways for each cell type as early molecular adjustments (3–48 h) based on their unique genomic structure. However, in the case of CisPt exposure we found each cell type eventually converged on similar pathways responses at later time points (72–96 h). We speculate that the two different immortalization processes could affect early transcriptional variations in the hTERT and LgT responses to CisPt. AFB1 toxicity in the LgT cells, compared to the absence of response in the hTERT cells, does suggest some important genomic differences to be further defined. We further expect that nephrotoxicant screening libraries will be valuable tools to uncover the unique capabilities of LgT and hTERT moRPTEC lines in various cellular formats including microfluidic RPTEC chips [14] and, 2D or 3D kidney in vitro models [92].

Single or repeated acute kidney injuries may lead to chronic kidney failure exacerbated by comorbidities, contributions by distal organs, therapeutics, and environmental exposures [93,94,95]. Notions of a “renal repair transcriptome” [44] and “conserved cellular responses to kidney injury” [96] have been forwarded along with regeneration by potential renal stem cells and tubular progenitors [97,98] to help explain transcriptional responses to kidney injury [43]. Single nucleus transcriptomics (snRNA-seq) can dissect out small populations of proximal tubule cells that fail to repair and persist after acute kidney injury. Cell populations harboring persistent injury may express proinflammatory and fibrotic genes that could lead to later clinical disease [96]. A recent snRNA-seq study of repeated low-dose CisPt treatment in mice at 9 weeks identified 16 different cell types and 17 cell clusters showing a unique proximal tubule injury and repair cluster [99]. We were encouraged to see upregulation of similar proinflammatory genes such as *Tnf*, *Il17*, and family members of CXC- and CC-motifs as found in our study.

## 4. Materials and Methods

Cell Immortalization and Culture: Mouse renal proximal tubule epithelial cells (moRPTECs) isolated from CD-1 mice were obtained commercially (Catalog #M4100; ScienCell, Carlsbad, CA, USA) and were transduced with lentivirus vectors containing either human TERT (hTERT) or SV40 Large T Ag (LgT) with antibiotic selection modules. The Ef1a_Large T-antigen Puro lentivirus was a gift from Linzhao Cheng [100] via Addgene (Watertown, MA, USA; Cat No. 18922) that contained a puromycin selection marker. The lentivirus expressing human telomerase reverse transcriptase (hTERT, NM_198253.2) was under enhanced expression using an EF1a promoter that contained a neomycin selection marker (GenTarget Inc., San Diego, CA, USA; Cat No. LVP1131-Neo).

RPTECs were grown on Biocoat collagen-1 coated 6-well plates (Corning, Corning, NY, USA; Cat No. 356400) in a commercial culture medium (ScienCell, Cat No. EpiCM-a) containing 2% FCS, Penicillin/Streptomycin antibiotics and a growth supplement. hTERT immortalized cells were selected and maintained by supplementing culture medium with 100 ng/mL G418-Geneticin™ (Invitrogen, Carlsbad, CA, USA; Cat No. 10131035) while LgT immortalized cells were selected and maintained at a concentration of 0.5 ug/mL of puromycin (Sigma-Aldrich, St Louis, MO, USA; Cat# 4512). Cells that had been passaged for over 50 cell divisions under continuous antibiotic selection with consistent cobblestone appearance and contact inhibition were considered stable cell lines. We observed that moRPTECs strongly adhered to culture plates surfaces and notably possessed cell self-adhesion properties in suspension. We found that successfully passaging moRPTECs required 1–2 PBS washes and 5–10 min incubation in PBS at 37 °C prior to trypsinization and 120× *g* centrifugation for 5 min to form a loose pellet, followed by resuspension in culture medium and plating.

Chemical Treatment: Aflatoxin B1 (AFB1; Sigma-Aldrich, Cat No. A6636) and Cisplatin (CisPt; Millipore-Sigma; Catalog No. 232120) were selected as representative nephrotoxicants to characterize transcriptional responses. Experiments were conducted in confluent cultures in 6-well plates. AFB1 was solubilized in DMSO (control) and CisPt was solubilized in phosphate buffered saline (PBS; control). Concentrations were chosen for minimal cytotoxicity at the highest level of chemical exposure in commercial growth media at 2 mL/well. Immortalized LgT and hTERT moRPTEC lines were grown to near confluence in 6-well tissue culture plates before treatment. To examine concentration-dependent transcriptional responses to CisPt or AFB1, cells were treated for 72 h with 0.25–2.5 μM CisPt, or with 12.5–100 μM AFB1. A second experiment determined transcriptional changes in LgT, and hTERT cells that were exposed to 2.5 μM CisPt over a 96 h time period. Samples at each time point were conducted in triplicate for CisPt and a triplicate time-matched control. At 0 h, untreated cells were nearly confluent and were harvested for RNA. After a single exposure to 2.5 μM CisPt, cells were harvested for RNA at 3 h, 6 h, or 24 h. For the remaining time points, media was changed daily containing either CisPt treatment or PBS at each 24 h period and then harvested for RNA 24 h later. Chemical treatments are summarized in Appendix A.

Mouse Cell Authentication: hTERT and LgT moRPTEC lines were submitted for mouse cell authentication using a short tandem repeat (STR) profiling (ATCC, Manassas VA; Cat No. 137-XV) performed according to published protocols [101]. Briefly, samples were analyzed for 18 mouse repeat (STR) loci including 2 markers to screen for the presence of human or primate species. Samples were processed with an ABI Prism^®^ 3500 xl Genetic Analyzer (ThermoFisher, Waltham, MA, USA). Authentication data are presented in Appendix A.

RNA and DNA isolation; RNA-seq: Cell viscosity was reduced by use of cell column shredders prior to nucleic acid isolation (Qiagen, Germantown, MD, USA; Qiashredder Cat No. 79654). For RNA isolation (Qiagen; RNeasy Mini, Cat No 74004), cells were washed with PBS prior to addition of lysis buffer and spin columns with on-column DNAase-1 digestion followed by washes, elution, and storage at −80 °C. RNA integrity was measured on an Agilent Model 5300 Fragment Analyzer (Agilent, Santa Clara, CA, USA) with RQN values of 9–10. RNA samples were prepared for RNA-seq by rRNA depletion, fragmentation (Covaris Inc., Woburn, MA, USA), bar coding, and library construction as previously described [102]. Briefly, pooled libraries were analyzed for cluster generation of 150 bp paired end fragments to produce a ~30X to 40X coverage of the mouse transcriptome using an Illumina NovaSeq instrument. Sequences were aligned to the mm10 mouse genome and analyzed for differential expression using DEseq2 [103]. Fastq data files were stored in the NCBI Sequence Read Archives (SRA) database under Bioproject PRJNA870295. DNA isolation (Qiagen DNeasy Kit, Cat No 69504) was performed for STR profiling for mouse cell authentication.

Nanostring Analysis: NanoString nCounter (NanoString, Seattle, WA, USA) was used as a multiplexed gene expression platform to test for specific transcripts as previously described [104]. Briefly, non-cross-reactive probes were custom synthesized and mixed to validate transcript expression after normalization with housekeeping transcripts (*Actb*; *Gapdh*, *Hprt*, *Rpl32*) that were determined by RNA-seq to be treatment stable. Samples were analyzed by nCounter at 150 ng RNA input. Data were analyzed by ANOVA and Newman–Keuls testing.

## 5. Future Directions

There are several considerations for future directions of the current work. Functional data for ion transporter activity and xenobiotic metabolic activation of these two cell lines would be important for placing context to the transcriptional changes that we reported. This could include functional assays for Oct (organic cation transporters, e.g., Oct1, Oct2, Oct3) and Oat (organic anion transporters, e.g., Oat1, Oat2, Oat3) transporters that are of clinical importance in the disposition of many drugs or xenobiotics. The creation of additional immortalized cell lines by human papilloma virus (HPV) or Epstein–Barr virus could provide additional breadth to the biological responses possible in moRPTEC lines. Further, the inclusion of different mouse strains of both sexes could add important gender representation and complexity of response to these two in vitro cell line models. A more comprehensive view of RPTEC responses to toxicant challenge could be gained by addition of other omics methodologies including proteomics, metabolomics, and epigenomics that could be strategically applied to critical early time points or more prolonged exposure paradigms to better simulate the conditions leading to chronic renal kidney disease. Finally, the value of in vitro findings and hypotheses generated from these renal cell lines would greatly benefit from *in vivo* testing in both animal models or more complex multicellular experimental constructs such as organoids or microphysiological systems. The development of interconnected, pulsatile culture media systems carrying signaling molecules and hormones from different cell types represent promising new technologies for combining different immortalized cell types to better understand the causes of toxicity and disease.

## 6. Conclusions

This study presents two new immortalized RPTEC lines with similar and some distinct capabilities in their response to two different nephrotoxicants. We believe the findings from these two cell lines can provide in vitro models for improved understanding of renal toxicity and repair mechanisms. First, data in the LgT cells suggest AFB1 may have direct nephrotoxic properties independent of hepatic metabolism and processing. Second, expression from CisPt experiments in both the cell lines support the concept of a ‘coordinated transcriptional program’ of injury signals and repair processes represented by (1) externalization of immune receptors as well as cytokine and chemokine secretion for leukocyte recruitment; (2) a robust expression of both synaptic and SAM molecules to facilitate increased expression of neural and hormonal receptors, ion channels/transporters, and trophic factors; and (3) expression of nephrogenesis transcription factors to assist in regeneration and repair. Many of the neural and cardiac canonical pathways evoked by pathway analysis are not unexpected since these tissues share similar cellular processes to kidneys in synaptic and bioelectric transmission, and intravesicular transport. Third, we anticipate differences in the immortalization processes of the parent RPTECs affect the genomics that shape the kinetics of phenotypic responses in these two cell lines—a research area that is actively being investigated in our laboratory. High throughput screening of LgT and hTERT-derived moRPTECs against chemical toxicant libraries will provide a wider perspective of biological responses and the unique capabilities of these cell lines in nephrotoxicity research.

## Figures and Tables

**Figure 1 ijms-24-14228-f001:**
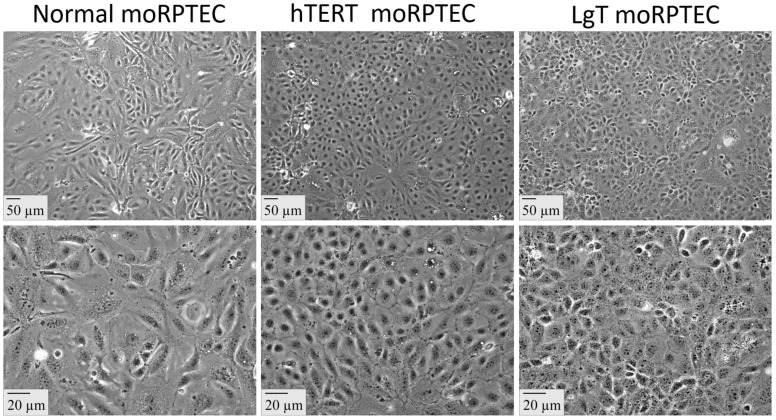
Cellular morphology of mouse renal proximal tubule epithelial cells (moRPTECs). Normal moRPTECs were immortalized with lentivirus vectors containing either human TERT (hTERT) or SV40 Large T antigen (LgT) with antibiotic selection modules. The top and bottom panels are at 10× and 40× magnifications, respectively, with scale bars in μm.

**Figure 2 ijms-24-14228-f002:**
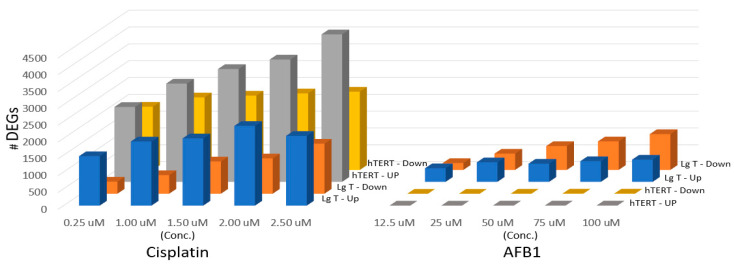
Differentially expressed genes (DEGs) with increasing concentrations of CisPt (Cisplatin) or AFB1 (aflatoxin B1) in moRPTECs. Cells were exposed daily to CisPt for 72 h and lysed for RNA isolation and subsequent RNA-seq analysis to determine differential expression (2× fold change, Adj *p* < 0.05).

**Figure 3 ijms-24-14228-f003:**
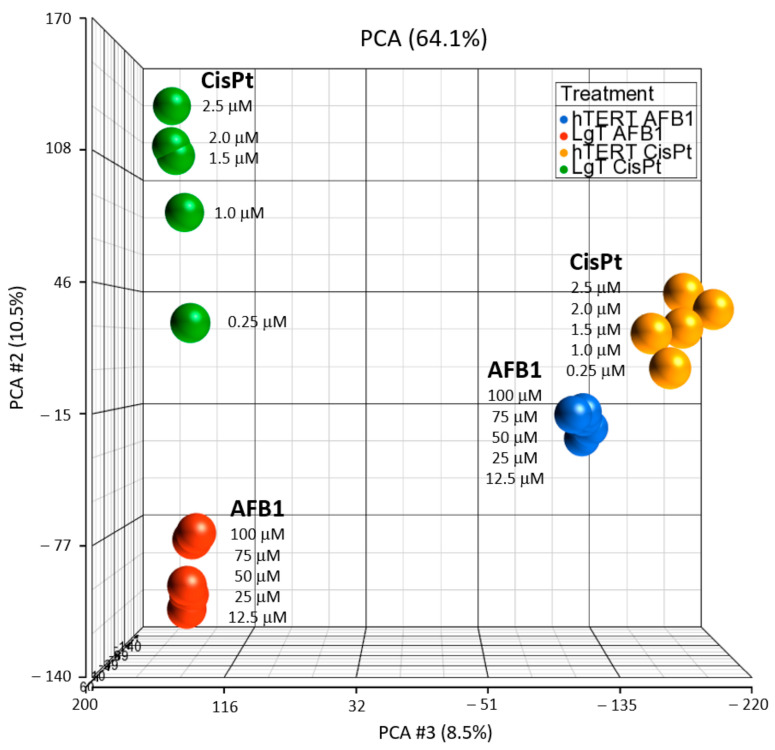
Principal component analysis of DEGs from 72 h exposure to CisPt or AFB1 in hTERT or Lg moRPTECs.

**Figure 4 ijms-24-14228-f004:**
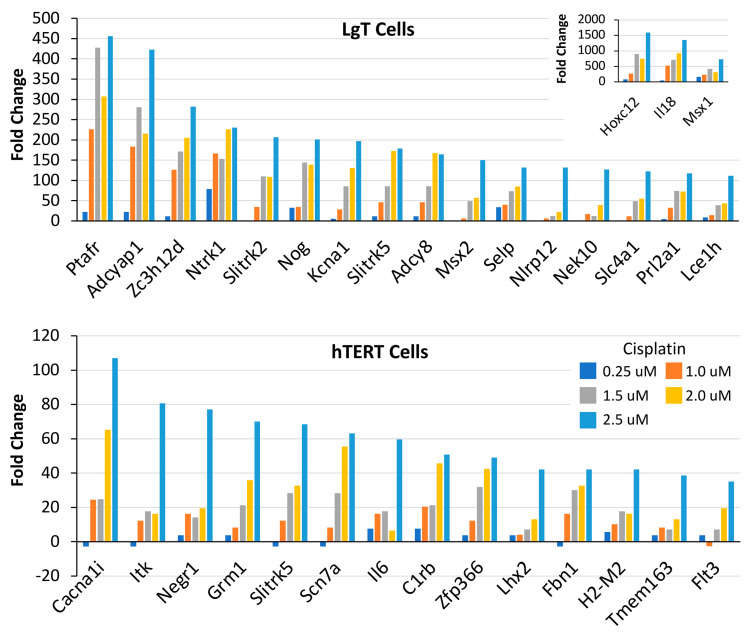
Concentration-related DEGs in LgT or hTERT moRPTECs. The top 100 DEGs were examined for concentration dependence after 72 h exposure to CisPt. The colored bar chart for CisPt concentrations refers to both upper and lower panels. Please refer to Abbreviations section for expanded gene descriptions.

**Figure 5 ijms-24-14228-f005:**
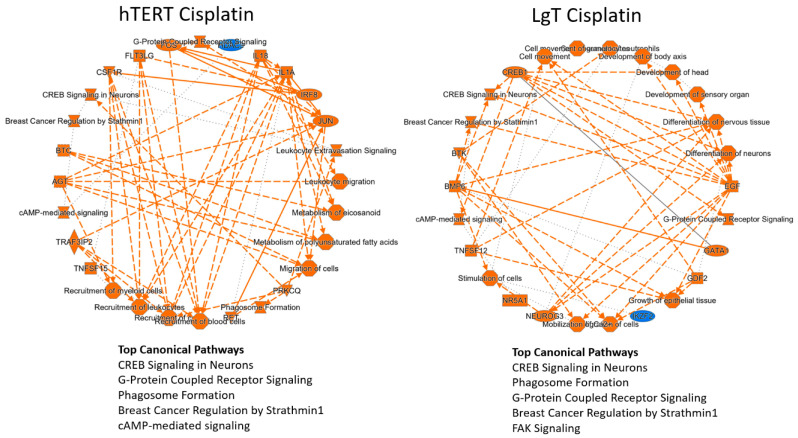
Activation of canonical pathways by CisPt. Pathway analysis was performed on DEGs in LgT or hTERT moRPTECs after 72 h exposure to CisPt.

**Figure 6 ijms-24-14228-f006:**
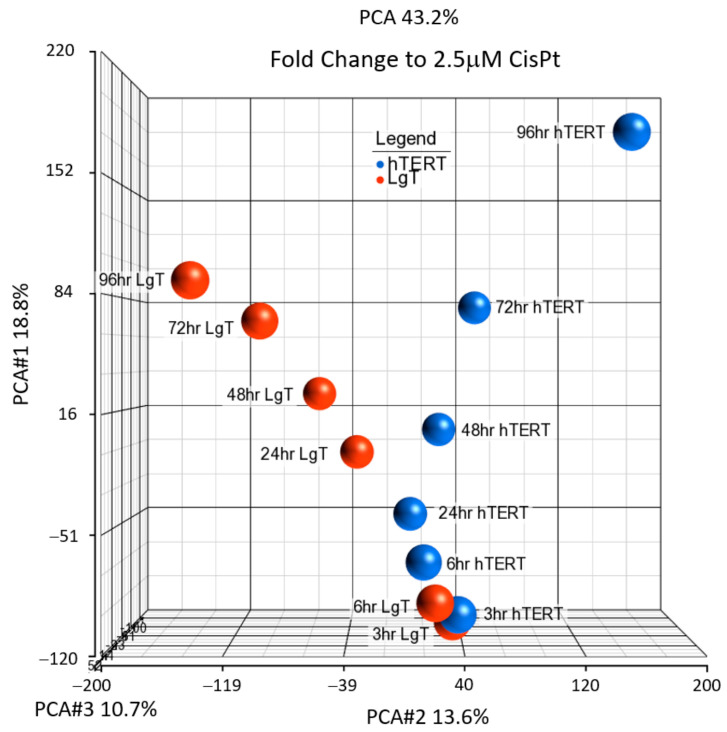
PCA analysis of DEGs in hTERT cells or LgT moRPTECs exposed to 2.5 µM CisPt over time. Cells were exposed to CisPt for 3, 6, or 24 h and then daily for 48, 72, or 96 h prior to RNA isolation for RNA-seq analysis.

**Figure 7 ijms-24-14228-f007:**
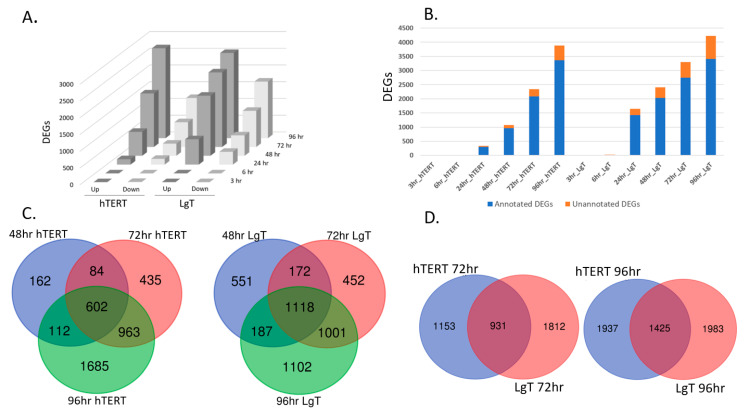
Differential expression in hTERT and LgT moRPTEC lines over time. Panel (**A**) shows differential expression (2X fold change, *p* ≤ 0.05) from 3 to 96 h. Panel (**B**) shows the proportion of annotated (blue) and unannotated genes (orange) after CisPt. Panel (**C**) is a Venn diagram of common and unique DEGs in hTERT or LgT cells at 48, 72, or 96 h after CisPt. Panel (**D**) show Venn diagram analysis of common and unique DEGs comparing hTERT to LgT DEGs after CisPt.

**Figure 8 ijms-24-14228-f008:**
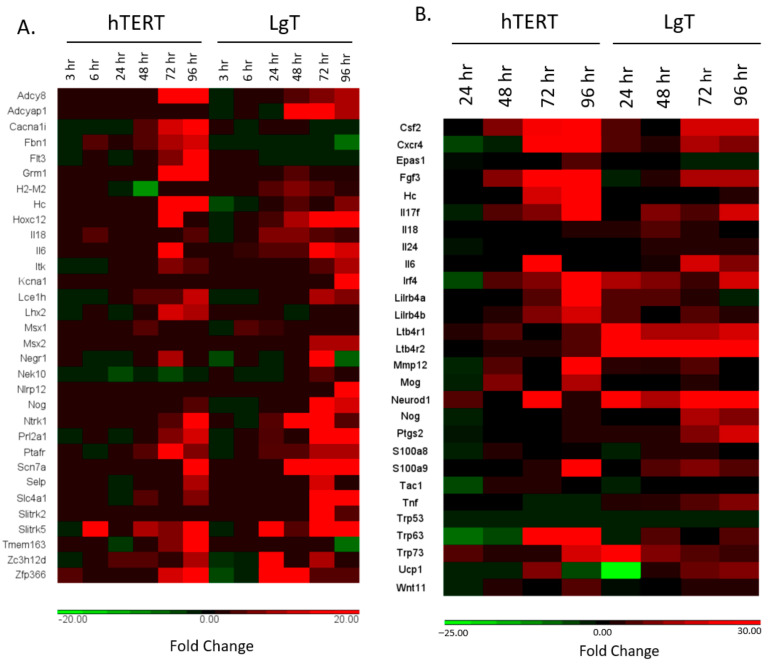
Heat maps of DEGs in hTERT and LgT moRPTECs. Panel (**A**) shows upregulated (red) and downregulated (green) DEGs, or no change in expression (black) from 3–96 h after CisPt. These genes were selected from the concentration–response experiment described in Figure 4. Panel (**B**) shows differential expression of upstream transcripts regulating pathway activation from 24–96 h after CisPt. DEGs are displayed as upregulated (red) and downregulated (green) DEGs, or no change in expression (black). See text and Abbreviations section for further details.

**Figure 9 ijms-24-14228-f009:**
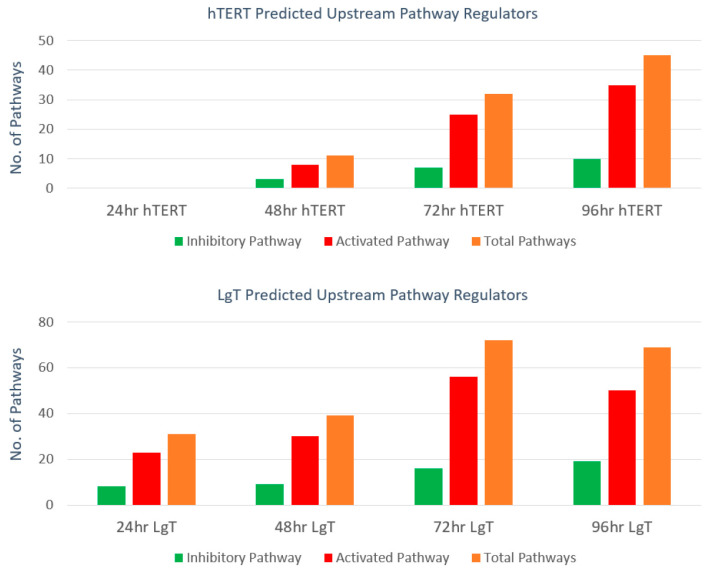
Number of upstream pathway regulators. Regulatory transcripts controlling downstream pathways were compiled according to activated, inhibitory, and total number of pathways for hTERT and LgT moRPTECs exposed to CisPt from 24 to 96 h.

**Figure 10 ijms-24-14228-f010:**
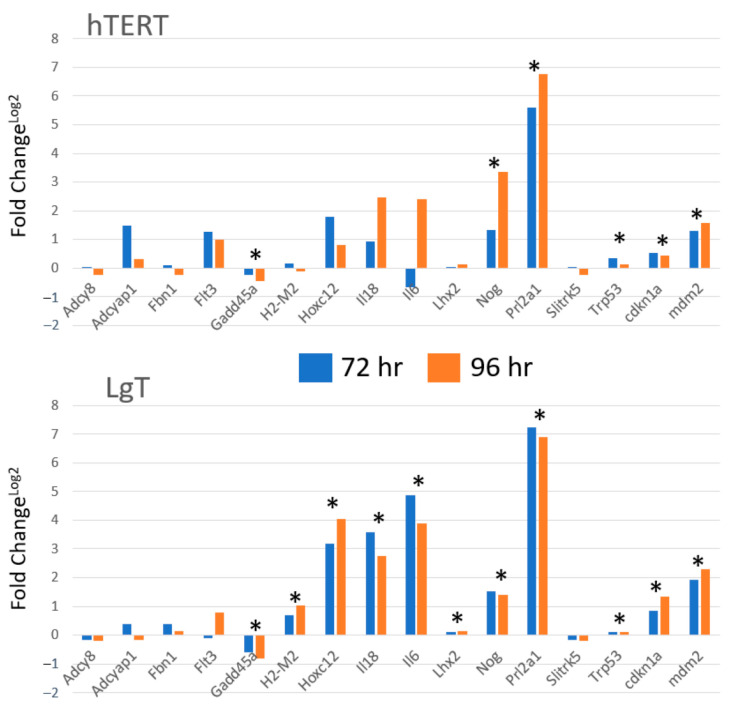
nCounter multiplexed analysis of transcripts in hTERT and LgT cells after 72 or 96 h CisPt exposure. Fold changes for each transcript were calculated by comparison of CisPt treatment to time matched controls. Data were analyzed by ANOVA and Newman–Keuls tests. (* indicates *p* < 0.05 significance). See Abbreviations section for gene descriptions.

**Table 1 ijms-24-14228-t001:** The highest twenty DEGs for hTERT and LgT in response to CisPt after 24–96 h. Highlighted DEGs have the following meanings: Red indicates an upregulated DEG with two or more occurrences in one cell type; Orange indicates an upregulated DEG that occurs in both cell types. Annotated genes are defined in the Abbreviations section.

Gene SYMBOL	24 h FC LgT	Gene SYMBOL	48 h FC LgT	Gene SYMBOL	72 h FC LgT	Gene SYMBOL	96 h FC LgT
*Shisal2b*	279.9	*Smim43*	222.6	*Exoc3l2*	231.6	*Ccdc184*	217.7
*Sult1b1*	261.7	*Abo*	189.0	*Neurog2*	145.9	*Dynap*	194.1
*Pla2g2f*	249.1	*Otof*	134.3	*Gm10634*	145.7	*Myh3*	184.5
*Hephl1*	239.6	*Cbln2*	133.1	*Shisal2b*	140.4	*Kcnd3*	183.4
*Rfx6*	235.8	*Robo4*	114.3	*Serpinb2*	129.5	*Nefl*	101.9
*Lhfpl1*	223.9	*Sparcl1*	102.2	*Ces2b*	127.9	*Mpped1*	91.1
*Gpr55*	220.0	*Lypd5*	101.0	*Pnliprp2*	127.7	*Lhx3*	87.2
*R3hdml*	191.9	*Gal*	99.8	*Stc1*	121.6	*Ptgs2os2*	86.4
*Wfikkn2*	149.8	*Col22a1*	97.7	*Tlr13*	119.2	*Acp4*	80.2
*Eef1a2*	137.8	*Tnnc2*	94.9	*Syt5*	115.5	*Chrna2*	78.6
*Ces2f*	134.6	*Gpr158*	89.0	*Nkx1-1*	115.4	*Jph2*	76.4
*Col6a3*	127.5	*Hephl1*	87.3	*Kprp*	103.0	*Il2rb*	73.0
*Sec14l3*	125.5	*Gm32742*	78.8	*Egr4*	99.5	*Mybpc1*	70.7
*Gsx1*	121.5	*Slc13a3*	77.9	*Dynap*	99.3	*Ifna13*	70.3
*Cnr2*	109.6	*Pramel16*	75.7	*Astl*	92.9	*Nfam1*	70.3
*Scn4b*	105.9	*Plxna4*	74.4	*Btnl10*	92.6	*Il10ra*	69.2
*Saa4*	94.9	*Hba-a1*	74.3	*Hhatl*	92.5	*Rab9b*	68.3
*Nme8*	94.5	*Pnpla1*	74.3	*Spta1*	85.1	*Krt90*	68.1
*Slitrk5*	90.9	*Ctsq*	74.1	*Abo*	78.5	*Ric3*	66.3
*Gm13652*	89.0	*Ces2e*	72.1	*Xirp2*	78.0	*Gast*	65.4
**Gene SYMBOL**	**24 h FC hTERT**	**Gene SYMBOL**	**48 h FC hTERT**	**Gene SYMBOL**	**72 h FC hTERT**	**Gene SYMBOL**	**96 h FC hTERT**
*Abo*	8.9	*Cacna1h*	178.3	*Otof*	239.0	*Tenm3*	287.9
*Ces2e*	7.5	*Ctsq*	143.4	*Ikzf3*	129.4	*Atp2b2*	276.8
*Apol8*	6.4	*Pla2g2f*	123.4	*Unc5a*	119.1	*Kcnn3*	270.8
*Cyyr1*	6.3	*Shisa2*	109.6	*Aox4*	114.2	*Abcc9*	220.1
*Spon2*	5.9	*Aknad1*	95.7	*Trim9*	111.3	*Otogl*	205.6
*Gm5737*	5.9	*Prss42*	92.4	*Mylk4*	111.2	*Mpped1*	198.9
*Des*	5.8	*Fam180a*	86.5	*Nyap2*	111.0	*Chrm3*	185.7
*Ces2f*	5.7	*Krt17*	84.3	*Calm4*	110.4	*Fibcd1*	182.8
*Inpp5d*	5.5	*Prss2*	79.0	*Cacna2d2*	109.9	*Sema5b*	182.6
*Exoc3l2*	5.4	*Adcyap1r1*	71.0	*Adamtsl3*	109.1	*Wdfy4*	178.2
*Ces2g*	5.3	*Cbln2*	67.6	*Mpped1*	109.0	*Cacna1s*	176.0
*Scn4b*	5.2	*Pmfbp1*	66.7	*Nppc*	106.6	*Trgc4*	174.2
*Gm7607*	5.1	*Otogl*	64.8	*Begain*	104.8	*Sparcl1*	171.5
*Ascl2*	4.9	*Nxph1*	64.7	*Prkcq*	104.4	*Ikzf3*	170.3
*Duox1*	4.8	*Nyap2*	64.1	*Pax7*	104.1	*Pde11a*	169.4
*2010310C07Rik*	4.7	*Colec10*	59.4	*Hpse2*	103.6	*Bank1*	167.5
*Podn*	4.6	*Rasgef1c*	59.3	*Slc5a4b*	101.7	*Sez6l*	158.0
*Zfp541*	4.5	*Vwa3b*	51.5	*Sv2c*	100.8	*Krtap3-2*	157.6
*Hic1*	4.3	*Kif28*	51.5	*Dsc1*	100.7	*Gabbr2*	154.6
*Fam83e*	4.2	*Foxl1*	51.1	*Dock2*	100.5	*Slc5a4b*	154.4

**Table 2 ijms-24-14228-t002:** The lowest twenty DEGs for hTERT and LgT in response to CisPt after 24–96 h. Highlighted DEGs have the following meanings: Green indicates a downregulated DEG with two or more occurrences in one cell type; Orange indicates a downregulated DEG that occurs in both cell types. Annotated genes are defined in the Abbreviations section.

Gene SYMBOL	24 h FC LgT	Gene SYMBOL	48 h FC LgT	Gene SYMBOL	72 h FC LgT	Gene SYMBOL	96 h FC LgT
*Enox1*	−9.7	*Elmo1*	−12.9	*Naaladl2*	−20.0	*Pacrg*	−38.1
*Cdh13*	−10.0	*Vrtn*	−13.2	*A830018L16Rik*	−20.9	*Clstn2*	−42.8
*Kirrel3*	−10.8	*Inpp4b*	−13.6	*Pcdhb9*	−22.4	*Syndig1*	−44.5
*Plcb1*	−10.9	*Pacrg*	−13.8	*AAdacl4fm3*	−22.6	*Naaladl2*	−45.4
*Aff3*	−11.7	*Stk32a*	−14.4	*Aadacl4fm5*	−24.0	*St6galnac3*	−46.0
*Afm*	−11.7	*Slc9a9*	−14.4	*Afm*	−24.8	*Slc13a1*	−46.9
*Gpc6*	−12.4	*Clstn2*	−14.5	*St6galnac3*	−26.2	*Fhit*	−55.2
*Sox5*	−12.4	*Msra*	−14.6	*Clstn2*	−26.2	*Klhl14*	−56.5
*Ccdc178*	−12.8	*Plcb1*	−14.8	*Tenm4*	−26.3	*Pard3b*	−57.1
*Slit3*	−13.2	*Aff2*	−16.6	*Pard3b*	−26.5	*Agmo*	−57.3
*Foxp2*	−13.4	*Gpc6*	−17.4	*Oog1*	−27.6	*Strit1*	−58.6
*Fhit*	−14.4	*Gm31814*	−17.6	*Aoc1*	−29.5	*Sox5*	−64.4
*Tenm4*	−16.4	*Pard3b*	−18.7	*Ptprq*	−31.8	*Erbb4*	−71.7
*Ecrg4*	−17.8	*Sox6*	−20.8	*Gpr88*	−40.6	*Tenm4*	−76.2
*Plppr1*	−22.8	*Ostn*	−20.9	*Dnajc5b*	−42.1	*Aff2*	−76.4
*Gm31814*	−27.1	*Ntrk3*	−32.6	*4930505A04Rik*	−44.3	*Aoc1*	−100.9
*Taf7l*	−35.6	*Gm24878*	−35.2	*Prr29*	−47.9	*Thsd7a*	−110.5
*Tff2*	−37.4	*Tenm4*	−39.5	*Ostn*	−53.6	*Dach1*	−150.9
*Kcnh8*	−37.8	*Fhit*	−47.2	*Gm17783*	−62.8	*Dnajc5b*	−157.2
*1810013D15Rik*	−41.3	*Ptprn2*	−97.3	*Fhit*	−80.7	*Gm17783*	−270.1
**Gene SYMBOL**	**24 h FC hTERT**	**Gene SYMBOL**	**48 h FC hTERT**	**Gene SYMBOL**	**72 FC hTERT**	**Gene SYMBOL**	**96 h FC hTERT**
*Tln2*	−4.8	*Pacrg*	−7.6	*D630024D03Rik*	−10.4	*Pacrg*	−22.7
*Tenm4*	−4.8	*Prkg1*	−7.7	*Acss1*	−10.8	*Cdh13*	−23.0
*Gabrb3*	−5.0	*Dnah11*	−7.8	*Cldn2*	−11.0	*Sox5*	−23.8
*Capn13*	−5.1	*Dab1*	−8.2	*Krtap1-5*	−11.0	*Npy2r*	−24.9
*Pclo*	−5.6	*Col5a2*	−8.8	*Gm13481*	−11.0	*Lrrc75b*	−25.3
*Msra*	−5.6	*Msra*	−9.0	*Slc17a1*	−11.1	*Sema5a*	−26.5
*Dcc*	−5.9	*Fhit*	−9.4	*Umod*	−11.3	*Fhit*	−27.1
*Sorcs1*	−6.4	*Gm47403*	−10.1	*A830018L16Rik*	−11.4	*Ggt1*	−27.8
*Adgrl3*	−6.6	*Cyp24a1*	−10.7	*Lrg1*	−12.4	*Cdh16*	−30.0
*Plcb1*	−7.0	*Angpt1*	−11.1	*Esrrg*	−12.5	*Fn3k*	−30.3
*Fhit*	−7.1	*Inpp4b*	−11.6	*Pacrg*	−13.4	*Gpc6*	−30.8
*Cyp24a1*	−7.1	*Thsd7b*	−12.7	*Fhit*	−13.9	*Sostdc1*	−32.2
*Cdh4*	−7.3	*Gpc6*	−13.9	*Gp6*	−16.5	*4930426D05Rik*	−33.3
*Inpp4b*	−7.5	*Cfap299*	−14.3	*Immp2l*	−16.7	*Immp2l*	−33.6
*Esrrg*	−7.7	*Cdh13*	−24.1	*Msra*	−16.7	*Umod*	−35.2
*Cdh13*	−7.8	*Immp2l*	−26.0	*Cfap58*	−20.3	*Cldn2*	−37.2
*Gpc6*	−8.5	*Ccdc7b*	−26.5	*Cdh13*	−27.6	*E130215H24Rik*	−41.0
*Thsd7b*	−9.6	*Spata22*	−33.3	*Gm22826*	−35.9	*4933431G14Rik*	−44.9
Sox5	−9.9	*Nobox*	−40.6	*Zscan4d*	−37.5	*A930003O13Rik*	−49.1
*Immp2l*	−10.1	*Egfl6*	−65.3	*Dnajc5b*	−49.6	*Hsd3b3*	−58.6

## Data Availability

Raw sequencing Fastq data files and project metadata have been deposited in the Sequence Read Archives database under Bioproject PRJNA870295.

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
