# Peer review of "Insights into Repeated Renal Injury Using RNA-Seq with Two New RPTEC Cell Lines"

_ijms, 2023, doi:10.3390/ijms241814228_

Round 1

Reviewer 1 Report

It was a real pleasure reading your manuscript, and I congratulate you for your hard work in collecting all these data. As you have already mentioned, the insidious onset of specific renal disorders could determine a late diagnosis and treatment initiation. Therefore, the study could represent a major step forward in highlighting the processes involved in the onset of several renal injuries, and consequently, on long-term, contributing to the development of new diagnostic tools and improving the current therapy management. The methodology and results were clearly described, and the conclusions were supported by your findings. I agree with your future direction that a thoroughly screening of LgT and hTERT could expand the research in this area of interest. One minor suggestion, please explain all the abbreviations from the abstract (i.e. hTERT, LgT).

Author Response

Response to Reviewer No. 1

We appreciate the generous comments of Reviewer No.1. The primary request was for a list of abbreviations.  We have gone through the manuscript text and Figures and now provide an alphabetized list of abbreviations at the end of the manuscript that includes organizations, programs and gene names. The list begins on line 648 of the Revised manuscript.

Reviewer 2 Report

This article describes the creation and characterization of two new immortalized mouse renal proximal tubule epithelial cell (moRPTEC) lines, one with human TERT (hTERT) and one with SV40 Large T antigen (LgT). The goal was to evaluate their utility as in vitro models for studying repeated low-level exposure to nephrotoxicants as a model of chronic kidney injury. The cells were treated with varying concentrations of cisplatin (CisPt) or aflatoxin B1 (AFB1) over time. RNA-seq analysis showed AFB1 elicited a transcriptional response only in LgT cells, suggesting direct nephrotoxicity. CisPt elicited robust transcriptional responses in both cell lines, with 2,000-5,000 differentially expressed genes indicating activation of injury, immune response, and repair pathways. Further time course RNA-seq analysis of CisPt exposure found both cell lines showed increasing differential gene expression over time, with the most changes at 72-96 hrs. Upstream pathway analysis predicted activation of inflammatory, immune cell recruitment, tissue remodeling, differentiation, and repair processes. About 25% of differentially expressed genes were shared between the two cell lines. The data supports the use of these new moRPTEC lines to study molecular mechanisms of repeated low-level chemical injury leading to chronic kidney disease. Their unique immortalization and response characteristics provide complementary models for mechanistic and therapeutic studies.

  1. Only two immortalization methods were tested (hTERT and SV40 LgT). Comparing additional methods like HPV E6/E7 or zinc finger nucleases could reveal further insights.
  2. A limited number of nephrotoxicants were evaluated. Testing a wider range of chemicals and concentrations would better characterize the cell lines' responses.
  3. No functional assays like transporter activity were performed. Adding functional data would provide mechanistic context to transcriptional changes.
  4. The in vitro conditions lack influences from systemic metabolism and circulating factors. Using 3D culture systems or microfluidics could better model in vivo conditions.
  5. Only male CD-1 mice were used as the cell source. Testing both sexes and additional strains could identify strain and sex specific differences.
  6. A limited time course was tested. Examining additional early and later time points could provide further temporal insight.
  7. Only RNA-seq was used for characterization. Adding proteomics, metabolomics, and epigenetic data would give a more complete view.
  8. Validation was limited. More extensive RT-qPCR confirmation of key transcripts would strengthen conclusions.
  9. Transfection with transporters like Oct2 could improve translation to human cells.
  10. In vivo correlation is needed. Xenografts in mice could determine if responses translate from in vitro findings.

Author Response

Response to Reviewer No. 2

We appreciate the comments provided by Reviewer No. 2.  There were 10 items for us to consider and I will address each.

  1. Comment: Other immortalization methods could reveal further insights.

Response: We agree and have included this point in a Future Directions section just before Conclusions.

  1. Comment: More nephrotoxicants would better characterize cell lines’ responses.

Response: We agree and have directly stated this point in the last sentence of the Conclusion section.

  1. Comment: Functional data would provide mechanistic context to transcriptional changes.

Response: This is an important point for which we agree.  We have included this point in the Future Directions section.

  1. Comment: In vitro conditions lack systemic metabolism and circulating factors, and using 3D culture or microfluidics could better model in vivo conditions.

Response: We agree on this important topic and have stated this point on lines 482-485.

  1. Comment: Only CD-1 mice were used as a cell source and additional strains with both sexes should be used.

Response: This is a great point by Review No.2 and we will make this point in the Future Directions section.

  1. Comment: A limited time course was tested and additional early and later time points could provide further insight.

Response: Yes, we agree on this point and will include it in the Future Directions section.

  1. Comment: Only RNA-seq was used and other Omics data would give a more complete view.

Response: We agree on this point and will provide this information in the Future Directions section.

  1. Comment: Validation was limited and more extensive analysis could strengthen the conclusions.

Response: We intend to perform more extensive validation of critical gene changes in future experiments when additional nephrotoxicants are screened.

  1. Comment: Transfection with transporters like Oct2 could improve translation to human cells.

Response: We agree that this would be an excellent set of experiments to perform in future work.

  1. Comment: In vivo correlation is needed such as xenografts in mice to translate in vitro findings.

Response: We agree with Review No.2 that there is much work to be performed for in vivo correlations of our in vitro findings, and will include this point in the Future Directions section (described below) and is now included in the revised manuscript (lines 500-520).

Future Directions

There are several considerations for future directions of the current work.  Functional data for ion transporter activity and xenobiotic metabolic activation of these two cell lines would be important for placing context to transcriptional changes that we report.  This could include functional assays for Oct (organic cation transporters, e.g. Oct1, Oct2, Oct3) and Oat (organic anion transporters, e.g. Oat1, Oat2, Oat3) transporters that are of clinical importance in the disposition of many drugs or xenobiotics.  The creation of additional immortalized cell lines by human papilloma virus (HPV) or Epstein-Barr virus could provide additional breadth to biological responses possible in RPTEC cell lines.  Further, the inclusion of different mouse strains of both sexes could add important gender representation and complexity of response to these two in vitro cell line models.  A more comprehensive view of RPTEC responses to toxicant challenge could be gained by addition of other omics methodologies including proteomics, metabolomics and epigenomics that could be strategically applied to critical early time points or more prolonged exposure paradigms to better simulate conditions leading chronic renal kidney disease. Finally, the value of in vitro findings and hypotheses generated from these renal cell lines would greatly benefit from in vivo testing in both animal models or more complex multicellular experimental constructs such as organoids or microphysiological systems. The development of interconnected, pulsatile culture media systems carrying signaling molecules and hormones from different cell types represent promising new technologies for combining different immortalized cell types to better understand causes of toxicity and disease.